# Rock Climbing Emergencies in the Austrian Alps: Injury Patterns, Risk Analysis and Preventive Measures

**DOI:** 10.3390/ijerph17207596

**Published:** 2020-10-19

**Authors:** Christopher Rugg, Laura Tiefenthaler, Simon Rauch, Hannes Gatterer, Peter Paal, Mathias Ströhle

**Affiliations:** 1Department of Anesthesiology and Critical Care Medicine, Medical University of Innsbruck, Anichstrasse 35, 6020 Innsbruck, Austria; christopher.rugg@tirol-kliniken.at (C.R.); laura.tiefenthaler@student.i-med.ac.at (L.T.); 2Institute for Mountain Emergency Medicine, EURAC Research, viale Druso 1, 39100 Bolzano, Italy; simon.rauch@eurac.edu (S.R.); hannes.gatterer@eurac.edu (H.G.); 3Department of Anaesthesiology and Intensive Care Medicine, “F. Tappeiner” Hospital, Via Rossini 5, 39012 Merano, Italy; 4Austrian Society for Alpine and High-altitude Medicine (ÖGAHM), Lehnrain 30a, 6414 Mieming, Austria; peter.paal@icloud.com; 5Department of Anaesthesiology and Intensive Care Medicine, Hospitallers Brothers Hospital, Paracelsus Medical University, Kajetanerplatz 1, 5010 Salzburg, Austria; 6Austrian Mountain Safety Board, 6020 Innsbruck, Austria

**Keywords:** rock climbing, emergencies, acute traumatic injuries, injury pattern, preventive measures, risk analysis

## Abstract

To elucidate patterns of and risk factors for acute traumatic injuries in climbers in need of professional rescue, a retrospective evaluation was performed of the Austrian National Registry of Mountain Accidents regarding rock climbing incidents over a 13-year timeframe from 2005 to 2018. From 2992 recorded incidents, 1469 were uninjured but in need of recovery, mainly when alpine climbing. Acute traumatic injuries (*n* = 1217) were often classified as severe (UIAA ≥ 3; *n* = 709), and commonly involved fractures (*n* = 566). Main injury causes were falls (*n* = 894) frequently preceded by rockfall (*n* = 229), a stumble (*n* = 146), a grip or foothold break-out (*n* = 143), or a belaying error (*n* = 138). In fatal cases (*n* = 140), multiple trauma (*n* = 105) or head injuries (*n* = 56) were most common, whereas lower extremity injuries (*n* = 357) were most common in severely injured patients. The risk for severe or fatal injuries increased with age and fall height when ascending or bouldering, during the morning hours, and when climbing without a helmet or rope. The case fatality rate was 4.7%, and the estimated total mortality rate was 0.003–0.007 per 1000 h of rock climbing. Acute traumatic injuries requiring professional rescue when rock climbing are often severe or fatal. Consequent use of a helmet when sport climbing, consistent use of a rope (particularly when ascending), proper spotting when bouldering, and proper training, as well as high vigilance when belaying are likely to help prevent such injuries.

## 1. Background

The popularity of rock climbing has grown steadily since it grew out of mountaineering during the last century. Since then, rock climbing has split into several categories, namely, alpine (traditional) climbing, ice climbing, sport climbing, and bouldering. Briefly describing these categories, alpine climbing is performed by placing and removing all necessary gear (e.g., nuts, cams, slings) for ascent and also for fall protection into a rock wall yourself. Ice climbing can be seen as a special subcategory of alpine climbing on inclined ice formations and extreme variants, such as free solo climbing, where climbers climb alone without belaying or aiding equipment. However, sport climbing and bouldering can be performed both indoors and outdoors. In outdoor sport climbing, permanent anchors and bolts are prefixed in close distance to the wall, suggesting higher levels of protection when used correctly. Additional protection can be provided by the use of crash pads for falls from a short distance. These pads are particularly useful when bouldering, which is performed without any equipment on small rock formations outdoors or on artificial walls at a low height indoors.

Rock climbing bears an inherent risk of injury and even death, yet the potential hazard levels vary considerably between categories [1]. When analyzing injury patterns related to rock climbing, differentiation between acute traumatic (e.g., fall from height, rockfall), atraumatic (supraphysiological loading), and chronic overuse injuries (e.g., pulley strain) seems wise. In alpine and ice climbing, higher risks for acute traumatic injuries have been described due to apparent dangers (e.g., falls, rockfall, technical failure) [2,3,4,5], while atraumatic and chronic overuse injuries seem to occur more often in sports climbing where increased protection may result in athletes pushing their limits [6,7,8]. A bulk of the recent literature has mainly focused on injuries reported in questionnaires or on climbers treated in an emergency department [3,7,9,10,11,12,13,14,15,16,17,18,19]. As acute traumatic injuries infrequently occur, the literature is rather scarce, and the more common overuse injuries are predominantly addressed. While overuse injuries are often related to forced and prolonged training and climbing intensity, accidents are mainly caused by errors, inattentiveness, or environmental influences [3]. The possible consequences range from merely getting lost or trapped in a rock wall, to minor injuries or even multiple trauma. To date, only a few studies have dealt with acute traumatic injuries requiring professional rescue during rock climbing [5,20,21]. A study from Grand Teton National Park, WY, USA, reported 144 rock climbing accidents in a 10-year period from 1970 to 1980 [20]. As every athlete had to register before entering the park, accurate incidence rates were calculated, and amounted to two accidents per 1000 climbers per year [20]. When referencing to this study, one must keep in mind that climbing techniques differed substantially compared to now. A further study reported 428 rock climbing accidents requiring rescue operations in Boulder County, CO, USA, over a 14-year period [21]. Moreover, a recent study using data from the International Alpine Trauma Registry reported 37 climbing accidents with multisystem trauma over 7.5 years across the Austrian, Italian, and Swiss regions [22]. While mainly focusing on climbing hot-spots [20,21,22], these studies do not allow for a comprehensive overview of rock climbing injuries requiring professional rescue on a national basis.

By evaluating the digital National Registry for Mountain Accidents, we were able to analyse rock climbing-related incidents in a 13-year timeframe from November 2005 to October 2018. The goal of this study was to obtain information on accident frequencies, mechanisms, preceding circumstances, and subsequent injury patterns and severity, and to perform a risk factor analysis in order to develop possible preventive measures.

## 2. Methods

The Ethics Committee of the Medical University of Innsbruck approved the study (AN2016-0018 358/4.8 390/5.5) and it was registered with Clinical Trials (NCT03755050). Anonymized data on emergencies and rescue operations in the Austrian Alps are collected by officers of the Alpine Police and stored in the digital National Registry for Mountain Accidents. As qualified alpinists, these officers have basic medical training (first aid and basic life support). As soon as a person requires a professional rescue, an entry is made in the registry. By 31 October 2018, this registry had grown to one of the largest registries for mountain accidents worldwide, containing more than 109,710 entries. Rock climbing-related accidents from initiation of the registry on 1 November 2005 to 31 October 2018 were extracted and analysed (Figure 1). 

Data obtained for this study included sex, age, nationality, date, time and weather conditions, performed climbing discipline (rock, sport, ice climbing, or bouldering) and activity (ascent, abseiling, descent, traverse) at the time of the emergency, injury severity, classification and localization, accident cause and consequences, and equipment used. Data are presented as median and interquartile range or count and percentage, as appropriate. Fisher’s exact test was performed to detect group differences in frequencies, and Mann Whitney’s U test for group differences of continuous data. By performing a logistic regression analysis, diverse independent variables were analyzed with regard to a possible prediction of an undesirable outcome, defined as a severe or fatal injury. The association was quantified by odds ratios (OR) adjusted for age and performed climbing discipline. In order to reduce family wise error rates and strengthen our analysis, we decided to rely on a 99% confidence interval (CI) concerning significance. Data were stored with Excel 2019 (Microsoft, Seattle, WA, USA) and processed with RStudio, version 1.2.5001, (RStudio Inc., Boston, MA, USA).

Due to regulations provided by Austrian law, documentation of injury severity initially took place according to the following definition: an injury is considered mild if the patient is able to resume work within 24 days, and severe if any fracture occurs except for a fracture of the nose, or the patient is unable to work for at least 24 days after the accident [23]. Aiming for comparability, we translated this definition into the following UIAA MedCom scores: uninjured—0; mild injury—1–2; severe injury—3–5; death on-site—6 [24]. As in-hospital data were missing to verify or further classify injury severities, more precise grading was not possible.

## 3. Results

### 3.1. Demographics and General Emergency Characteristics

A total of *n*= 2992 rock climbing accidents were analyzed. Of these, 78.7% involved male persons, and 21.3% female. The median age was 33 (26–45) years. While causes of emergencies and their subsequent consequences are illustrated in Table 1, injury characteristics are shown in Table 2. Emergencies were concentrated on the weekends, with 52.6% of all cases occurring on Saturdays or Sundays. The median time of day was noon (10:00–15:00 o’clock) and the predominant weather condition was sunny (65.9%), followed by cloudy (12.4%). Emergencies were recorded all year long, but the main reporting season was summer, followed by fall (Table 3). 

The leading climbing discipline performed when incurring an emergency was alpine climbing, followed by sport climbing, ice climbing, and bouldering (Table 3). Emergencies occurred predominantly during the ascent, followed by abseiling, descent, and traverse (Table 3).

### 3.2. Injury Severity and Localization

Nearly half of all persons involved remained uninjured (Table 2). Of these, 82.7% were performing alpine climbing, and the most common causes of an emergency were being lost or trapped (*n* = 474), being an uninjured companion, unable to continue climbing (*n* = 441), or exhaustion (*n* = 172). When injuries occurred, the majority were severe, and mostly involved the ankle (*n* = 181; 25.5%) or the head (*n* = 110; 15.5%; Figure 2). Additionally, injuries to the lower leg (*n* = 61; 8.6%) and the foot (*n* = 69; 9.7%) were more pronounced. The main causes of severe injuries were rockfall (*n* = 94), grip or foothold break-out (*n* = 91), or a belaying error (*n* = 81). This mainly led to a fall with rope impact or a fall with ground impact, and predominantly resulted in fractures (*n* = 523; 73.8%). Mild injuries were more equally distributed, but with peaks in the head (22.3%) or multiple-located injuries (17.1%; Figure 2). Here, rockfall (*n* = 92) and stumbling and falling (*n* = 36) accounted for nearly half of all known causes, and primarily led to contusions, strains, sprains (*n* = 130) or wounds, lesions, and lacerations (*n* = 115). Fatalities occurred in 4.7% of all cases. The main causes leading to death were injuries affecting multiple localizations (*n* = 71; 50.7%) or the head (*n* = 56; 40.0%; Figure 2). Three quarters of the 140 deaths (*n* = 105; 75.0%) were classified as a multiple trauma, and *n* = 110 occurred after a fall with ground impact. 

### 3.3. Risk Factors for Undesirable Outcome

Undesirable outcomes, defined as severe or fatal injury, were present in 28.4% (*n* = 849) of all persons involved in a rock climbing emergency, and were primarily the result of a fall with rope impact (*n* = 311), a fall with ground impact (*n* = 310), or direct injury (*n* = 163) caused by preceding stumbling and falling (*n* = 135), grip or foothold break-out (*n* = 102), rockfall (*n* = 99), or belaying error (*n* = 96). 

Table 3 gives an overview of the odds of undesirable outcomes for different risk factors, and Table A1 displays the injury localization, causes of injury, direct consequences, performed activity, and the type of discipline leading to an undesirable outcome for every risk factor.

Undesirable outcomes increased from age 30, peaking in those over 70 years old (Table 3). Belay errors were more common with younger age (<25 years: 17.3%; ≥25 years: 10.2%; *p* = 0.024), while stumbles and falls were very common over all ages (Table A1).

The proportions of emergencies leading to an undesirable outcome were 25.2% during alpine climbing, 41.9% during sport climbing, 44.4% during ice climbing, and 54.5% during bouldering (Table 3). During alpine climbing, higher mortality but a lower fraction of severe injury was observed. Furthermore, 55.1% of the persons rescued when alpine climbing were uninjured, blocked as the uninjured companion, or otherwise lost or trapped. In other climbing subdisciplines, the percentage of uninjured persons was 30.6% for ice climbing, 23.9% for sport climbing, and 3.0% for bouldering. Predominant causes leading to severe or fatal injury were a grip or foothold break-out in alpine climbing, a belaying error in sport climbing, avalanches in ice climbing, and stumbles and falls when bouldering (Table A1).

In general, ascending was the most common activity, followed by abseiling, descending, or traversing (Table 3). Regarding undesirable outcomes, 68% occurred during the ascent, followed by abseiling (14.5%), descent (8.4%), and traversing (1.9%; Table A1). Severe injury occurred most frequently when ascending, and relative mortality was highest when traversing, followed by descending, ascending, or abseiling (Table 3). Being uninjured was most common when abseiling or descending.

In general, helmets were usually worn and ropes usually used when involved in a rock climbing emergency (Table 3). Severe or fatal injuries were more probable when not wearing a helmet (OR 1.81; CI 1.35–2.43), with the main injury localization being the head (Table 3 and Table A1). In comparison to all other subgroups, sports climbing predominated. Stumbling and falling or a belaying error were the main preceding causes commonly leading to falls with ground impact (Table A1). The increase in relative mortality was pronounced when climbing rope-free, and therefore also was the risk for undesirable outcomes (OR 1.67; CI 1.35–2.22; Table 3). Bouldering or descending showed a significantly higher percentage in the rope-free group, but ascending while performing alpine climbing was the predominant activity in both groups (Table A1). Stumbling and falling, as well as a grip or foothold break-out were the primary causes, and multiple traumas from a fall with ground impact was the most frequent result.

## 4. Discussion

The present study contributes to a more detailed picture of rock-climbing injuries by systematically analyzing country-wide incidents that required a professional rescue. Almost half of all persons involved in rock-climbing emergencies remained uninjured, but if an injury occurred, the consequences were often severe, frequently involving fractures. In general, mortality was low and acute traumatic injuries ending fatally were mostly multiple trauma or head injuries, whereas lower extremity injuries, especially of the ankle, predominated in severe injuries. Falling after a preceding stumble, grip, or foothold break-out or belaying error were the primary accident causes. With regard to rock climbing-related incidents requiring a professional rescue, odds ratios for severe or fatal injuries increased with age, height of fall, while bouldering or ascending, during the morning hours, and when climbing without a helmet or rope. Particularly with regard to the latter two, severe or fatal injuries while climbing without a helmet or rope mainly occurred while ascending in sport climbing concerning helmets, and while ascending in alpine climbing concerning rope usage. Consistent use of a helmet, also when sport climbing, consistent use of a rope, particularly on a traditional ascent, proper training in and high vigilance while belaying, proper spotting when bouldering, situational awareness, especially with regard to personal fitness, and environmental conditions are likely to help prevent poor outcomes.

### 4.1. Demographics and General Emergency Characteristics

In relation to the most recent study analyzing rock climbing rescue operations, the general emergency characteristics in our study were comparable with a clear male predominance and a concentration on weekends in summer, fall, and spring, between noon and early evening [21]. There also, nearly half the people involved remained uninjured, but were in need of extrication [21]. When injured, they were also injured rather severely, falls were the most common cause, and the lower extremities or the head were affected most.

### 4.2. Injury Severity and Localization

At first glance, the reported rather severe injury pattern in this study stands in contrast to that of other studies reporting on a preferred upper extremity involvement [7,13,16,19,25,26,27,28]. Differentiating more accurately reveals that acute traumatic injuries often involve the lower, while atraumatic or chronic overuse injuries predominantly involve the upper extremities [4,5,7,8,12,16,20,25,27,28,29,30]. Of course, overuse injuries are not restricted to the upper extremities, just as traumatic injuries are not restricted to the lower extremities [31,32]. Survey studies among active climbers show a predominance of minor injuries, but there are inconsistent results with regard to the distribution of injury types. Results range from almost equal percentages of overuse, acute atraumatic, and acute traumatic injuries to a proportion as high as 93% for overuse and only 7% for traumatic injuries [6,7,14,16,18,19]. Clearly hindering comparability are differences in patients included, depending on the study design (surveys, questionnaires, in-hospital, or rescue data). With regard to the rescue data presented in this study, we must assume that victims with milder and/or upper limb injuries were rather capable of self-rescue than those with more severe injuries or lower limb affection. This is supported by the high proportion of fractures, lower limb involvement, and the severely injured which were reported by us, and is in line with other comparable studies [21].

### 4.3. Risk Factors for Undesirable Outcome

Risk factor analyses have been done before, but usually rely on data from surveys and questionnaires, again including many minor and chronic overuse injuries. Reported risk factors for severe or fatal injuries in this study, however, must all be seen in the context of a preconditional requirement of professional rescue.

In previous studies, older rock climbers demonstrated a higher proportion of overuse injuries, especially degenerative conditions, and also an increase in injury severity when injured but no increase in acute injury rate [14,17,18,33]. Statements on injury rates cannot be done with the existing data in this study, but the effect of age on injury severity is confirmed. Present data show a clear age-dependent increase in the fraction with a worse outcome. 

With respect to climbing disciplines, previous studies have stated that the injury risk during alpine climbing is greater than that for other disciplines [3,8,14]. A questionnaire study from Sweden concluded that bouldering was associated with an increased risk, but the majority of persons suffered overuse injuries [7]. In our analyses, when requiring professional rescue, bouldering was the discipline with the highest risk for severe injuries, even though no mortality was recorded. An assumed higher proportion of viable self-rescue in bouldering certainly influences this ratio. Ice climbing is also considered to be more dangerous than other disciplines due to the presence of objective and external dangers [3]. However, in our analysis, ice climbing had a risk comparable to outdoor sport climbing. Grip or foothold break-out was the leading cause of undesirable outcomes in alpine climbing, a belaying error in sport climbing, avalanches in ice climbing, and stumbling or falling in bouldering. This is of special interest with regard to injury prevention, as training in the belay technique and careful tour preparation might be able to prevent at least some of these incidents. While bouldering, spotting the climber and correct positioning of high-quality crash pads may help to reduce injuries.

Only a few studies have analyzed the specific activity performed prior to injury occurrence, and therefore only little is known about risks for severe or fatal injuries in this regard. Alpine or sport leading has been shown to be associated with higher injury rates [13,14,18,19]. The present study was able to show that not only ascending, but also traversing had a higher mortality rate. Situational unawareness and risk underestimation might play a role when traversing. Incidents during abseiling and descending were significantly less severe, although they also involved the risk of falling with ground impact and subsequent death.

The use of a helmet is known to prevent undesirable outcomes, though not only in rock climbing. It is appalling enough that in certain areas, up to 80% of climbers practice rock climbing without a helmet [25,28]. Only 18% of all cases in this study involved persons not wearing a helmet, but the percentage increased to 28% in the group with undesirable outcomes. Particularly regarding outdoor sport climbing, 58% were not wearing a helmet prior to an emergency. Of them, 43% were injured severely, and 3% fatally. The present data remind us that consistent use of a helmet also when performing outdoor sport climbing and acute vigilance when belaying can help prevent an unfortunate outcome. 

Similarly, rope use must be seen as a very effective preventive measure. Whether rope-free climbing is performed by professional athletes in the form of free solo tours or by recreational climbers simply underestimating or possibly even seeking risks, falling when not using a rope almost always leads to severe or fatal injuries [6]. Interesting are the circumstances prior to the emergency, as getting stuck or merely performing bouldering may also lead to emergencies classified as rope-free. Despite a higher proportion of bouldering, the main activity practiced before suffering an undesirable outcome when climbing rope-free was ascending in alpine climbing.

Although most emergencies occurred in the afternoon, the greatest risk for adverse outcomes was experienced in the morning. Internal factors, like lack of a proper warm-up, or still insufficient alertness, or also external factors like morning dew, for example, may have contributed to this outcome. Apart from that, the percentage of those lost and uninjured but in need of help was clearly higher in the evening and at night.

Importantly, the height of the fall was linked to adverse outcomes in this study, with a mortality rate as high as 37.5% for falls over 20 m. As inevitable as this correlation seems, there have been inconsistent findings in the past, with some studies confirming, and others denying this connection [34,35,36,37].

In general, a fatal outcome was almost exclusively a result of multiple trauma or head injury after a fall with ground impact, as also described in other studies [5,21]. The overall case fatality rate was rather low, namely 4.7%. This is comparable to similar studies of climbing-related rescue operations in specific regions like Boulder County (5.5%), but is higher than that for other sports (canyoning 1.9%, via-ferrata climbing 3.7%) [21,38,39]. Mortality rates from 18% to 28% have also been reported in previous climbing studies [4,20]. 

### 4.4. Overall Mortality Estimation

A recent 5-year analysis of the national emergency department sample in the United States showed that over 99% of climbing-associated mortalities occur before arrival at the hospital [40]. Therefore, the 140 observed deaths in our 13-year timeframe should be an exact estimation of rock climbing-related fatalities in Austria. In order to further estimate an overall mortality rate, injury rates must be known. Two independent survey studies have reported similar rates for the occurrence of acute traumatic injuries during rock climbing. The first study excluded nontraumatic injuries and published a rate of 0.2 injuries per 1000 h of climbing [14]. The second study indicated an injury rate of 4.4/1000 h of climbing, with traumatic injuries amounting to 0.3/1000 h [7]. 

Importantly, not all injuries require professional rescue. Data from Yosemite National Park state that 27% of those injured while climbing were in need of professional rescue [5]. In a study of canyoning-related injuries from our department, merely 16% required rescue prior to presenting at the emergency department [38]. Traumatic injuries requiring professional rescue can therefore be estimated at 0.03–0.08/1000 h of climbing. Taking the 1523 people injured in this study into account, an estimated 20–50 million hours of rock climbing and a mortality rate of 0.003–0.007/1000 h of climbing can be calculated. This is comparable to mortality rates of 0.02/1000 h in similarly conducted studies of canyoning, and 1 per 10 million hours of via-ferrata climbing [38,39].

### 4.5. Limitations

Our results must be interpreted with respect to the preconditional requirement of professional rescue. It must be assumed that victims with mild and/or upper limb injuries are capable of rescuing themselves, and thus disregarded. While these missing data are clearly a limitation of the study, the chosen design offers the opportunity to specifically distinguish risk factors leading to severe or fatal injuries, which can be seen as highly relevant from a clinical and preventive point of view. Apart from that, we refrained from analyzing sex differences throughout the manuscript, as merely 21.3% of the climbers were female, and therefore total numbers were low. Additional limitations are the retrospective design of this study and missing clinical data on pre-hospital reported injury patterns.

## 5. Conclusions

Although general mortality is low and the risk of being in need of rescue from a mountainous environment while uninjured is high, acute traumatic injuries requiring professional rescue when rock climbing are often severe or fatal. Consequent use of a helmet when sport climbing, consistent use of a rope (particularly when ascending), proper spotting when bouldering, and proper training, as well as high vigilance when belaying may help prevent such injuries.

## Figures and Tables

**Figure 1 ijerph-17-07596-f001:**
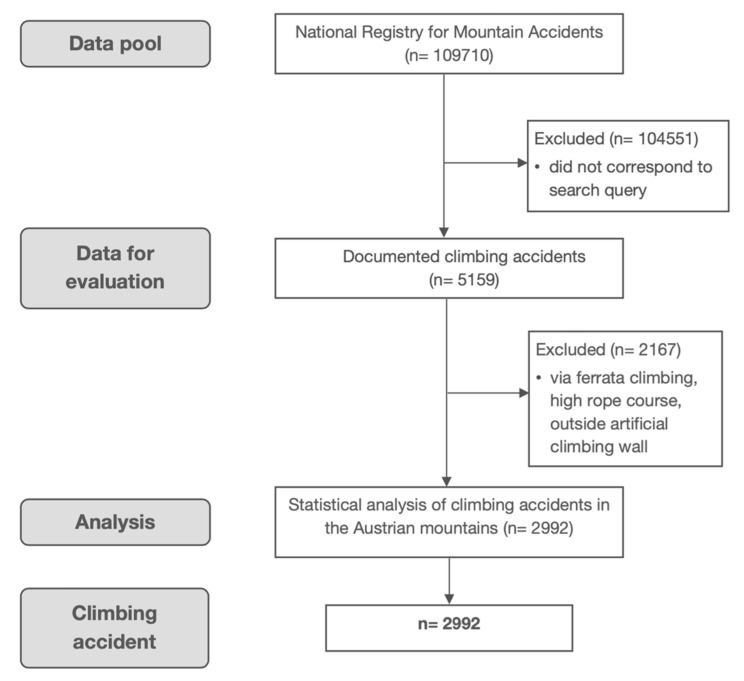
Consolidated Standards of Reporting Trials (CONSORT) flow chart.

**Figure 2 ijerph-17-07596-f002:**
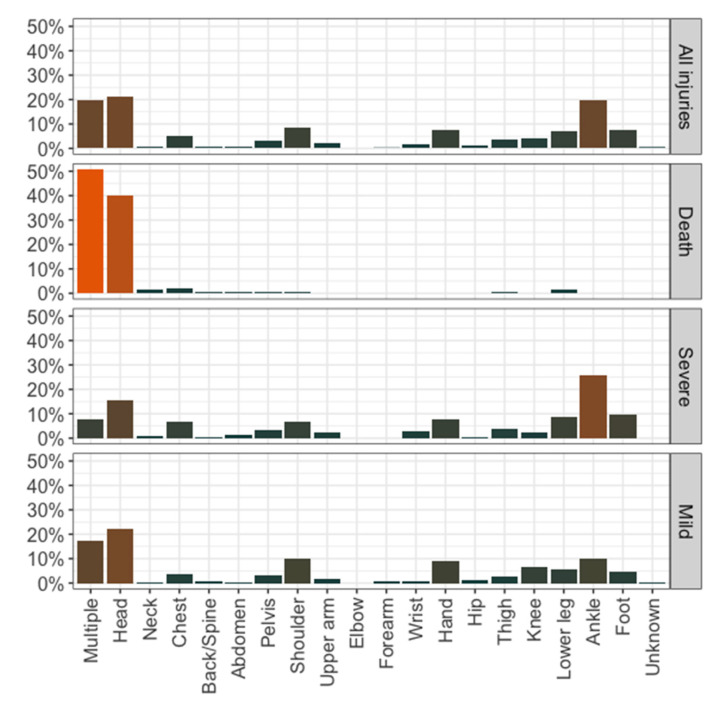
Injury localization depending on injury severity.

**Table 1 ijerph-17-07596-t001:** Characteristics of rock climbing emergencies in Austria from 2005 to 2018, ranked according to frequency.

Emergency Characteristics (*n* = 2992)	Classification	*n* (%)
Cause of emergency	Unknown	556 (18.6)
	Lost or trapped	517 (17.3)
	Uninjured companion *	505 (16.9)
	Rockfall	229 (7.7)
	Exhaustion	187 (6.3)
	Stumble and fall	146 (4.9)
	Grip or foothold break-out	143 (4.8)
	Belaying error	138 (4.6)
	Abseiling error	121 (4.0)
	False alarm	45 (1.5)
	Other	405 (13.5)
Direct consequence of preceding cause	Blockage	1004 (33.6)
	None	550 (18.4)
	Fall with rope impact	497 (16.6)
	Direct injury	429 (14.3)
	Fall with ground impact	379 (12.7)
	Unknown	115 (3.8)
	Fall of entire rope team	18 (0.6)

* Unable to continue climbing and therefore classified as an emergency.

**Table 2 ijerph-17-07596-t002:** Injury characteristics of rock climbing emergencies in Austria from 2005 to 2018.

Injury Characteristics (*n*= 2992)	Classification		*n* (%)
Injury severity	Uninjured	UIAA MedCom 0	1469 (49.1)
	Mild	UIAA MedCom 1–2	368 (12.3)
	Severe	UIAA MedCom 3–5	709 (23.7)
	Dead	UIAA MedCom 6	140 (4.7)
	Unknown	Unknown	306 (10.2)
Emergency classification (1578 entries in *n* = 1450 patients)	Fracture		566 (35.9)
	Contusion/Strain/Sprain		264 (16.7)
	Wound/Lesion/Laceration		250 (15.8)
	Multiple trauma		150 (9.5)
	Luxation		97 (6.1)
	Exhaustion		48 (3.0)
	Concussion		47 (3.0)
	Hypothermia		46 (2.9)
	Internal injury		33 (2.1)
	Unknown		32 (2.0)
	Burn injury		12 (0.8)
	Cardiovascular disorder		12 (0.8)
	Stroke		1 (0.1)
	Suffocation		1 (0.1)
Injury localization (1503 entries in *n* = 1315 patients)	Multiple		257 (19.5)
	Head		279 (21.2)
	Neck		11 (0.8)
	Chest		68 (5.2)
	Back/Spine		5 (0.4)
	Abdomen		11 (0.8)
	Pelvis		40 (3.0)
	Shoulder		114 (8.7)
	Upper arm		26 (2.0)
	Elbow		0 (0)
	Forearm		2 (0.2)
	Wrist		23 (1.7)
	Hand		101 (7.7)
	Hip		13 (1.0)
	Thigh		46 (3.5)
	Knee		53 (4.0)
	Lower leg		91 (6.9)
	Ankle		259 (19.7)
	Foot		97 (7.4)
	Unknown		7 (0.2)

**Table 3 ijerph-17-07596-t003:** Odds ratios adjusted for age and the performed climbing discipline, calculated by logistic regression, with regard to a severe or fatal injury.

Category		Total (*n*)	Severely Injured (%)	Fatally Injured (%)	Odds Ratio
					(99% CI)
**Age group ^§^**	***18–29 yrs.*** ***^#^***	1028	19.6	3.3	
	***<18 yrs.***	93	20.4	3.2	0.97 (0.50–1.90)
	***30–49 yrs*** **.**	1371	26.5	2.9	**1.48 (1.15–1.89)**
	***50–69 yrs.***	447	25.7	11.6	**2.22 (1.61–3.06)**
	***>70 yrs.***	35	22.9	31.4	**4.52 (1.83–11.13)**
**Climbing discipline ^+^**	***Alpine ^#^***	2207	20.2	5	
	***Sport***	394	39.6	2.3	**2.33 (1.73–3.13)**
	***Ice***	36	41.7	2.8	**2.56 (1.07–6.17)**
	***Bouldering***	33	54.5	0	**4.16 (1.66–10.42)**
**Activity**	***Ascent ^#^***	1792	27.7	4.6	
	***Abseiling***	506	20.9	3.4	**0.65 (0.48–0.89)**
	***Descent***	331	15.1	6.3	**0.59 (0.40–0.86)**
	**Traverse**	46	26.1	8.7	1.11 (0.48–2.54)
**Use of helmet**	***With helmet ^#^***	2363	21.5	3.7	
	***Without helmet***	549	33.2	8.7	**1.81 (1.35–2.43)**
**Use of rope**	***With rope ^#^***	2410	24.1	2.4	
	***Without rope***	502	23.5	15.7	**1.67 (1.35–2.22)**
**Time of emergency**	***12:00–16:00 ^#^***	1461	28.2	5.2	
	***17:00–21:00***	808	10.4	1.9	**0.30 (0.22–0.41)**
	***22:0006:00***	88	9.1	1.1	**0.26 (0.10–0.65)**
	***07:00–11:00***	619	32,6	7.8	**1.39 (1.07–1.81)**
**Height of fall**	***<5 m ^#^***	85	50.6	0	
	***5–9 m***	111	72.1	0	**2.67 (1.20–5.94)**
	***10–19 m***	101	65.3	7.9	**3.07 (1.32–7.12)**
	***>20 m***	88	48.9	37.5	**7.52 (2.72–20.74)**

^§^ Adjusted for the performed climbing discipline only. ^+^ Adjusted for age only. ***^#^*** Reference for odds ratio. Odds ratio in bold type indicates statistical significance.

## Data Availability

No data are available. Participant data from the Digital National Registry for Mountain Accidents hosted by Österreichisches Kuratorium für Alpine Sicherheit- Austrian Mountain Safety Board. All data are deidentified. The data set was delivered containing only serial numbers for each participant. Protocol and statistical analysis plans are available.

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
