# Peer review of "Rock Climbing Emergencies in the Austrian Alps: Injury Patterns, Risk Analysis and Preventive Measures"

_ijerph, 2020, doi:10.3390/ijerph17207596_

Round 1
Reviewer 1 Report
This in an interesting study where the authors aimed to elucidate injury patterns of and risk factors for acute traumatic injuries in climbers in need of professional rescue and therefrom to develop preventive measures. The aim of this study is very important, because of common traumatick injuries in rock climbing sports. Unfortunately the authors presented only retrospective, descriptive data of injured people without any practical implications or usefull conclusions. This study in this form has very low scientific validity. The authors should rewrite and strengthen the rationale at the introduction , and what is more importand to discuss their data more comprehensively. The practical implications of these results ( from large population !!!) should be clearly explained based on authors own results and other similar studies.
Author Response
Dear Reviewer,
We would like to truly thank you for your time and your efforts in critically working on our manuscript!
We took your comments very seriously and believe that the resulting changes immensely added to the quality of our manuscript.
Adoptions made and sentences added can be tracked via Words “track-changes” within the manuscript!
Please find below a summary of the major changes done:
ABSTRACT:
The abstract was adopted. Results were presented more precisely. Consequences rephrased with special regard to existing results!
“Abstract: To elucidate patterns of and risk factors for acute traumatic injuries in climbers in need of professional rescue a retrospective evaluation was performed of the Austrian National Registry of Mountain Accidents regarding rock climbing incidents over a 13-year timeframe from 2005 to 2018. From 2,992 recorded incidents 1,469 were uninjured but in need for recovery mainly when alpine climbing. Acute traumatic injuries (n=1,217) were often classified as severe (UIAA≥3; n=709) and commonly involved fractures (n=566). Main injury causes were falls (n=894) frequently preceded by rockfall (n=229), a stumble (n=146), a grip or foothold break-out (n=143) or a belaying error (n=138). In fatal cases (n=140), multiple trauma (n=105) and/or head injuries (n=56) were most common, whereas lower extremity injuries (n=357) were most common in severely injured patients. The risk for severe or fatal injuries increased with age and fall height, when ascending or bouldering, during the morning hours and when climbing without a helmet or rope. The case fatality rate was 4.7% and the estimated total mortality rate was 0.003–0.007 per 1000 hours of rock climbing. Acute traumatic injuries requiring professional rescue when rock climbing are often severe or fatal. Consequent use of a helmet also when sport climbing, consistent use of a rope, particularly when ascending, proper spotting when bouldering and proper training as well as high vigilance when belaying are likely to help prevent such injuries.”
INTRODUCTION:
The introduction was strengthened by adding explanations of rock climbing subdisciplines and clarifying why our data is of special interest:
Line 39ff.:
“The popularity of rock climbing has grown steadily since it grew out of mountaineering during the last century. Since then, rock climbing has split into several categories, namely alpine (traditional) climbing, ice climbing, sport climbing and bouldering. Briefly describing these categories, alpine climbing is performed by placing and removing all gear necessary (e.g. nuts, cams, slings) for ascent and also fall protection into a rock wall yourself. Ice climbing can be seen as a special subcategory of alpine climbing on inclined ice formations and extreme variants, such as free solo climbing where climbers climb alone without belaying and aiding equipment also. Sport climbing and bouldering however can be performed in- and outdoors. In outdoor sport climbing permanent anchors and bolts are prefixed in close distance to the wall suggesting a higher protection when used correctly. Additional protection can be provided by the use of crash pads for falls from short distance. These pads are particularly useful when bouldering, which is performed without any equipment on small rock formations outdoors or on artificial walls in low height indoors.
Rock climbing bears an inherent risk of injury and even death, yet the potential hazard levels vary considerably between categories.[1] When analyzing rock climbing related injury patterns, differentiation between acute traumatic (e.g. fall from height, rockfall), atraumatic (supraphysiological loading) and chronic overuse injuries (e.g. pulley strain) seems wise. In alpine and ice climbing higher risks for acute traumatic injuries have been described due to apparent dangers (e.g. falls, rockfall, technical failure) [2–5] while atraumatic and chronic overuse injuries seem to occur more often in sports climbing where increased protection may result in athletes pushing their limits.[6–8] A bulk of recent literature mainly focuses on injuries reported in questionnaires or on climbers treated in an emergency department.[3,7,9–19] As acute traumatic injuries infrequently occur, literature is rather scarce and the more common overuse injuries are predominantly addressed…“
METHODS:
Logistic regression analysis was adopted to a significant confidence interval of 99%. The effect of the season on severe or fatal injuries therefore missed significance. We deleted this section from results and discussion.
Furthermore, a better classification of injury severity with regard to comparability took place. We added the following:
Line 179ff:
“Due to regulations provided by Austrian law, documentation of injury severity initially took place according to the following definition: an injury is considered mild if the patient is able to resume work within 24 days and severe if any fracture, except a fracture of the nose, occurs or the patient is unable to work for at least 24 days after the accident.[23] Aiming for comparability we translated this definition into the following UIAA MedCom Scores: uninjured – 0; mild injury – 1-2; severe injury – 3-5; death on-site – 6.[24] As in-hospital data was missing to verify or further classify injury severities a more precise grading was not possible.“
RESULTS:
As mentioned above a classification for injury severity (UIAA MedCom) was added and logistic regression adopted to a 99% confidence interval.
DISCUSSION:
First and foremost, the first paragraph of the discussion was rewritten in order to clarify the importance of our findings!
Line 411ff: “The present study contributes to the overall picture of rock-climbing injuries by systematically analyzing country-wide incidents that required professional rescue.
Almost half of the people who were involved in rock climbing emergencies remained uninjured, but if an injury occurred, the consequences were often severe, frequently involving fractures. In general, mortality was low and acute traumatic injuries ending fatally were mostly multiple trauma or head injuries, whereas lower extremity – especially ankle – injuries predominated in severe injuries. Falling after a preceding stumble, grip or foothold break-out or belaying error were the primary accident causes. With regard to rock climbing related incidents requiring professional rescue, odds ratios for severe or fatal injuries increased with age, height of fall, while bouldering or ascending, during the morning hours and when climbing without a helmet or rope. Particularly with regard to the latter two, severe or fatal injuries while climbing without a helmet or rope mainly occurred while ascending in sport climbing concerning helmet and while ascending in alpine climbing concerning rope usage. Consistent use of a helmet, also when sport climbing, consistent use of a rope, particularly on a traditional ascent, proper training in and high vigilance while belaying, proper spotting when bouldering, situational awareness, especially with regard to personal fitness and environmental conditions are likely to help prevent poor outcome.”
Within the discussion of 4.1. Demographics and general emergency characteristics we added the following:
Line 431: “When injured, they were also injured rather severely, falls were the most common cause and the lower extremity, or the head were affected most.”
Line 501: “Clearly hindering comparability are differences in patients included depending on the study design (surveys, questionnaires, in-hospital or rescue data). With regards to the rescue data presented in this study, we must assume that victims with milder and/or upper limb injuries were rather capable of self-rescue than those with more severe injuries or lower limb affection. This is supported by the high proportion of fractures, lower limb involvement and severely injured reported by us and is in line with other comparable studies.[21]”
Line 509: “Risk factor analyses have been done before, but usually rely on data from surveys and questionnaires again including many minor and chronic overuse injuries. Reported risk factors for severe or fatal injuries in this study, however, must all be seen in context of preconditional requirement of professional rescue.
In previous studies, older rock climbers demonstrated a higher proportion of overuse injuries, especially degenerative conditions and also an increase in injury severity when injured but no increase in acute injury rate.[14,17,18,33] Statements on injury rates cannot be done with the existing data in this study but the effect of age on injury severity is confirmed. Present data show a clear age-dependent increase in the fraction with worse outcome.
Line 545: “Particularly regarding outdoor sport climbing 58% were not wearing a helmet prior to the emergency. Of them, 43% were injured severely and 3% fatally. The present data remind us that consistent use of a helmet also when performing outdoor sport climbing and acute vigilance when belaying can help prevent an unfortunate outcome.“
CONCLUSIONS:
We shortened and clarified our conclusions with special regard to the presented results!
“Although general mortality is low and the risk being in need for rescue from a mountainous environment uninjured high, acute traumatic injuries requiring professional rescue when rock climbing are often severe or fatal. Consequent use of a helmet also when sport climbing, consistent use of a rope, particularly when ascending, proper spotting when bouldering and proper training as well as high vigilance when belaying may help prevent such injuries.”
Reviewer 2 Report
This is important research that needs to be shared with members of the climbing, rescue and emergency services community. In general the manuscript would be improved by explaining many terms and some methodology for a broader audience that are neither climbers not statisticians. Numerous minor suggestions and requests for clarification are provided below.
19. The word "Background" should be replaced with "Purpose" or "Objective" to match the sentence that follows. I also recommend removing the word "injury "to avoid word repetition, i.e. "To elucidate patterns of and risks factors for…".
21-21. "severely affected" is vague and perhaps redundant with "acute traumatic injuries".
23-24. The second sentence following Results reads like a truism. Everyone knows that stumbles and hold breaks lead to falls. I believe there is a bigger point that author intends to convey though. I recommend making "falls" the subject of the sentence and ending with a statement about the frequency due to stumbles, hold breaks or belay errors.
29. Delete the first sentence after "Conclusions". I don't believe "acute traumatic injuries" are "accompanied by" severe or fatal injuries, I believe they "are" the severe or fatal injuries and poor outcomes.
29 – 36 Why is there no mention in the abstract of the top three causes of emergencies which collectively make up over half of all emergencies? Based on the analyses performed and the results presented, this was not an analysis of climbing injuries exclusively, but rather an analysis of all climbing emergencies logged in the National Registry for Mountain Accidents during the 13-year time period and excluding certain disciplines (via ferrata, plastic, etc.). Highlighting the proportion of emergencies due to being lost or trapped would send an important message to alpinists about the hazards they are most likely to encounter and should prepare for, which is stated as a goal of the study on line 64.
41. I suggest splitting the term "rope climbing" into traditional rock climbing, alpine climbing and sport climbing and developing this paragraph further with very short explanations of each discipline. Sport, alpine and traditional rock climbing represent very different albeit overlapping categories of climbing with very different risk and injury profiles. Ice climbing is also performed with a rope (by most), which means the reader may develop the wrong understanding of the difference between "rope climbing" and "ice climbing" if the sentence is left as is. Sport climbing is mentioned on line 76 and 100, and alpine climbing is mentioned on line 99, but these terms have not been defined for the reader.
81. State more directly why logistic regression analysis was performed. "regarding undesirable outcome" does not explain exactly what logistic regression estimates and many readers will not have the statistical background to infer the purpose or interpret the results.
Table 1. Uninjured companion as a category may need to be explained as this term would also describe anyone in a climbing group that is uninjured and not involved in an emergency. What about this uninjured companion is the cause of an emergency?
- The caption for table 3 is misleading because table 3 shows the adjusted odds ratios calculated by logistic regression, but does not show the "Odds of undesirable outcomes".
Table 3 – The calculation of 21 odds ratios, each held to a significance level of .05, gives a family wise error rate of .64. In other words, a 64% exists that the analysis will produce at least one type 1 error (false positive), even if in reality no meaningful differences exist. Obviously, you have many significant odds ratios, but I suggest using 99% confidence intervals to reduce the familywise error rate and increase reader confidence in your outcomes.
129. "fall rope impact" --> fall with rope impact ( I recommend using this structure throughout the manuscript)
135-136. "An accentuation of belaying error as the cause". Awkward! Suggestion: "Belay errors were more common with younger age (<25 years: 17.3%).
Author Response
Dear Reviewer,
we would like to thank you for your time and efforts in helping us improve our manuscript. We gladly integrated your suggestions.
Please find below point by point answers to your requests.
- The word "Background" should be replaced with "Purpose" or "Objective" to match the sentence that follows. I also recommend removing the word "injury "to avoid word repetition, i.e. "To elucidate patterns of and risks factors for…".
Due to editing guidelines from the publisher, sub-headlines in the abstract were completely removed and the sentences rephrased. We were glad to adopt the mentioned sentence according to your recommendation. Thank you!
Line 19: „To elucidate patterns of and risk factors for acute traumatic injuries in climbers in need of professional rescue a retrospective evaluation was performed of the Austrian National Registry of Mountain Accidents regarding rock climbing incidents over a 13-year timeframe from 2005 to 2018“
21-21. "severely affected" is vague and perhaps redundant with "acute traumatic injuries".
We do agree this phrasing was not very clear. Although of course often severe, acute traumatic injuries do not necessarily have to be. Especially when using this term to differentiate between acute traumatic, atraumatic and chronic overuse injuries also mild injuries can be included.
Also due to other recommendations we decided to rephrase this part of the abstract as follows:
Line 21: “Acute traumatic injuries (n=1,217) were often classified as severe (UIAA≥3; n=709) and commonly involved fractures (n=566).“
23-24. The second sentence following Results reads like a truism. Everyone knows that stumbles and hold breaks lead to falls. I believe there is a bigger point that author intends to convey though. I recommend making "falls" the subject of the sentence and ending with a statement about the frequency due to stumbles, hold breaks or belay errors.
We gladly changed the sentence as follows:
Line 24: “Main injury causes were falls (n=894) frequently preceded by rockfall (n=229), a stumble (n=146), a grip or foothold break-out (n=143) or a belaying error (n=138).“
- Delete the first sentence after "Conclusions". I don't believe "acute traumatic injuries" are "accompanied by" severe or fatal injuries, I believe they "are" the severe or fatal injuries and poor outcomes.
We deleted this part and rephrased the sentence as follows:
Line 30: “Acute traumatic injuries requiring professional rescue when rock climbing are often severe or fatal.“
29 – 36 Why is there no mention in the abstract of the top three causes of emergencies which collectively make up over half of all emergencies? Based on the analyses performed and the results presented, this was not an analysis of climbing injuries exclusively, but rather an analysis of all climbing emergencies logged in the National Registry for Mountain Accidents during the 13-year time period and excluding certain disciplines (via ferrata, plastic, etc.). Highlighting the proportion of emergencies due to being lost or trapped would send an important message to alpinists about the hazards they are most likely to encounter and should prepare for, which is stated as a goal of the study on line 64.
Thank you once more for your critical reception of our manuscript!
As we set focus on the analysis of acute traumatic injuries while rock-climbing we Initially decided against inclusion of data on uninjured in the abstract. We must agree though that it is a main finding of our study, and better elucidates the entire picture after including these statements. As you mentioned this information is of importance to the alpinist with regard to most likely to encounter hazards.
The following sentences were included:
Line 22: “From 2,992 recorded incidents 1,469 were uninjured but in need for recovery mainly when alpine climbing.“
Also, in the first paragraph of the Discussion following sentences were adopted:
Line 423f: “Almost half of the people who were involved in rock climbing emergencies remained uninjured, but if an injury occurred, the consequences were often severe, frequently involving fractures.“
And the conclusion was rephrased:
Line 626f: “Although general mortality is low and the risk being in need for rescue from a mountainous environment uninjured high, acute traumatic injuries requiring professional rescue when rock climbing are often severe or fatal.”
- I suggest splitting the term "rope climbing" into traditional rock climbing, alpine climbing and sport climbing and developing this paragraph further with very short explanations of each discipline. Sport, alpine and traditional rock climbing represent very different albeit overlapping categories of climbing with very different risk and injury profiles. Ice climbing is also performed with a rope (by most), which means the reader may develop the wrong understanding of the difference between "rope climbing" and "ice climbing" if the sentence is left as is. Sport climbing is mentioned on line 76 and 100, and alpine climbing is mentioned on line 99, but these terms have not been defined for the reader.
We totally agree with your objection!
We expanded the introduction by more precise definitions of the climbing disciplines of interest!
The introduction was rephrased to the following:
Line 39fff:
“The popularity of rock climbing has grown steadily since it grew out of mountaineering during the last century. Since then, rock climbing has split into several categories, namely alpine (traditional) climbing, ice climbing, sport climbing and bouldering. Briefly describing these categories, alpine climbing is performed by placing and removing all gear necessary (e.g. nuts, cams, slings) for ascent and also fall protection into a rock wall yourself. Ice climbing can be seen as a special subcategory of alpine climbing on inclined ice formations and extreme variants, such as free solo climbing where climbers climb alone without belaying and aiding equipment also. Sport climbing and bouldering however can be performed in- and outdoors. In outdoor sport climbing permanent anchors and bolts are prefixed in close distance to the wall suggesting a higher protection when used correctly. Additional protection can be provided by the use of crash pads for falls from short distance. These pads are particularly useful when bouldering, which is performed without any equipment on small rock formations outdoors or on artificial walls in low height indoors.”
- State more directly why logistic regression analysis was performed. "regarding undesirable outcome" does not explain exactly what logistic regression estimates and many readers will not have the statistical background to infer the purpose or interpret the results.
As we feel that reaching as many readers as possible is of importance, we must thank you for your precise reading and great tips:
We changed the section as follows:
Line 173: “. By performing a logistic regression analysis, diverse independent variables were analyzed with regard to a possible prediction of an undesirable outcome, defined as severe or fatal injury. The association was quantified by odds ratios (OR) adjusted for age and performed climbing discipline. In order to reduce family wise error rates and strengthen our analysis we decided to rely on a 99% confidence interval (CI) concerning significance.“
Table 1. Uninjured companion as a category may need to be explained as this term would also describe anyone in a climbing group that is uninjured and not involved in an emergency. What about this uninjured companion is the cause of an emergency?
Dear Reviewer, we must agree that the uninjured companion is not easily comprehensible.
The following sentence is added to the results 3.2 Injury severity and localization:
Line 221: „Nearly half of all persons involved remained uninjured (Table 2). Of them, 82.7% were performing alpine climbing and the most common causes of an emergency were being lost or trapped (n= 474), being an uninjured companion, unable to continue climbing (n= 441) or exhaustion (n= 172).“
We decided to also add the following as footnote beneath Table 1: „*unable to continue climbing and therefore classified as emergency“
- The caption for table 3 is misleading because table 3 shows the adjusted odds ratios calculated by logistic regression, but does not show the "Odds of undesirable outcomes".
Agree!
We changed as follows:
Table 3 header: “Odds ratios adjusted for age and performed climbing discipline, calculated by logistic regression, with regard to a severe or fatal injury.“
Table 3 – The calculation of 21 odds ratios, each held to a significance level of .05, gives a family wise error rate of .64. In other words, a 64% exists that the analysis will produce at least one type 1 error (false positive), even if in reality no meaningful differences exist. Obviously, you have many significant odds ratios, but I suggest using 99% confidence intervals to reduce the familywise error rate and increase reader confidence in your outcomes.
We once more must truly thank you for your help!
We decided to do as recommended and utilize a 99% confidence interval with regard to significance.
In doing so all but one significant odds ratio remained. The factor spring (season of the emergency) missed significance after the adoptions. For better clarity and readability, we therefore removed this section (season) from the analysis.
- "fall rope impact" --> fall with rope impact ( I recommend using this structure throughout the manuscript)
Corrections made!
Thank you!
135-136. "An accentuation of belaying error as the cause". Awkward! Suggestion: "Belay errors were more common with younger age (<25 years: 17.3%).
Agree. Not phrased very well!
We rephrased as suggested and would like to thank you once more for your help!
Line 373: “Belay errors were more common with younger age (<25 years: 17.3%; ≥25 years: 10.2%; p = 0.024), while stumbles and falls were very common over all ages (Table A1).”
Reviewer 3 Report
Overall:
The study covers important questions and data, nevertheless a few points are not addressed adaequately. E.g. a common injury classification is not used etc… see the points below.
In specific:
Abstract
- 22 regarding rock climbing incidents over a 13-year timeframe
please state which years
- step break-out
the term step is not commonly used in rock climbing discriptions. The correct term should be foot hold.
- undesirable outcome
what is an undesirable outcome, a fatality? A permanent disability?
In results no injury grading is presented. Please present grading according to a common used classification, best the UIAA medcom classification.
Conclusion:
- Acute traumatic injuries when
- 30 rock climbing are often accompanied by poor outcome in terms of severe or fatal injuries.
No, most acute injuries in all other studies are not severe, while severe injuries are still happening. Nevertheless most injuries are ankle fractures etc and not severe.
Introduction
- The existing literature mainly focuses on injuries reported
- 46 in questionnaires or on climbers treated in an emergency department.[2–14] In both cases overuse
- 47 injuries caused by an excessive climbing load are predominantly addressed.
This statement is false for the studies involving Emergerncy departements only, eg. Buzzacot and Nelson. These NEISS data exclude overuse injuries.
- While overuse injuries
- 48 are often related to forced and prolonged training intensity,
should be training and climbing intensity
- A study from Grand Teton National Park, WY, USA, reported 144 rock climbing accidents in a 10-
- 53 year period from 1970 to 1980.[15]
This study is good, but very old. The climbing techniques in the 80ies vary from today, please address this.
Methods
- Severity of injury was defined according to Austrian law: an injury was considered mild if the
- 86 patient was able to resume work within 24 days and severe if any fracture, except a fracture of the
- 87 nose, occurred or the patient was unable to work for at least 24 days after the accident.[19
I would strongly suggest to use a standard classification of trauma severity. I recommend the UIAA medical commission score as the int. climbing medicine bodies classification.
step break-out
please check with a native speaker if this term is correct. I am not a native English speaker but it seems unusual to me and I know the term foothold is mostly used.
Results
Table 2 : exhaustion is not an injury but a medical condition, same for hypothermia. You may need to rephrase injury into injuries and medical conditions, same for stroke etc.
Injury localization (1503 entries in n= 1315 patients)
Also here we would recoomend to use a classification, e.g. OSICS for localization.
Table 3
18 – 29 yrs.# < 18 yrs.
Why don’t you start chronologically, with under 18 then 18-256 etc..?
Climbing discipline:
How do you differ from alpine to sport? Please define
- Helmets were usually worn and ropes usually used when involved in a rock climbing
- 154 emergency (Table 3).
also in sport climbing??
Discussion
- Consistent use of a helmet, also when sport climbing,
- 173 consistent use of a rope, particularly on an ascent, high vigilance while belaying, including proper
- 174 training in belaying, thorough tour preparation and situational awareness, especially with regard to
- 175 personal fitness and environmental conditions, can help prevent poor outcome.
even if logical it can be stated for sure as not proofed scientifically. I would suggest change to are likely or may protect
- The use of a helmet is known to prevent undesirable outcomes, not only in rock climbing.
- 216 Appalling enough, that in certain areas up to 80% of climbers practice rock climbing without a
- 217 [20,23] Only 18% of all cases in this study involved persons not wearing a helmet, but the
- 218 percentage increased to 28% in the group with undesirable outcome. The present data remind us that
- 219 consistent use of a helmet also when performing sport climbing and acute vigilance when belaying
- 220 can help prevent an unfortunate outcome.
well I certainly disagree for sport climbing and other studies on sport climbing injuries showed very little head injuries… this needs to be discussed.
- Spring was the season with the highest rate of severe or fatal injuries. The most plausible
- 234 explanation for this phenomenon is that overall fitness was probably somewhat poorer in spring than
- 235 in the summer or fall.
also: in winter rock freezes, then I taws, and rock fall is always higher in spring!
Tables and Figures
Conclusion:
- Conclusions
- 276 In rock climbing accidents requiring professional rescue, the main causes of death and severe
- 277 injuries were falls after a stumble, grip or step break-out or a belaying error. Consequences were
- 278 multiple trauma or head injuries in the fatal cases and lower extremity injuries in the severely injured.
- 279 The risk for such undesirable outcomes increased with age and height of fall, when ascending or
- 280 bouldering, in spring, during the morning hours and when climbing without a helmet or rope. The
- 281 case fatality rate was 4.7% and overall mortality was estimated at 0.003–0.007 per 1000 h of rock
- 282 Preventive measures can be consistent use of a helmet also when sport climbing, consistent
- 283 rope use particularly when ascending, acute vigilance when belaying and proper belay training,
- 284 thorough tour preparation and situational awareness, especially concerning personal fitness and
- 285 environmental conditions. While bouldering, spotting the climber and correct positioning of high-
- 286 quality crash pads may help to reduce injuries.
this is the same as the abstract! Should actually draw a conclusion, be more precise and shorter
Author Response
Overall:
The study covers important questions and data, nevertheless a few points are not addressed adaequately. E.g. a common injury classification is not used etc… see the points below.
Dear Reviewer,
We would like to truly thank you for your valuable input!
Please find below point by point answers to your requests:
In specific:
Abstract
22 regarding rock climbing incidents over a 13-year timeframe
please state which years
Thank you for your attentiveness!
We changed the line as follows:
Line 19: “To elucidate patterns of and risk factors for acute traumatic injuries in climbers in need of professional rescue a retrospective evaluation was performed of the Austrian National Registry of Mountain Accidents regarding rock climbing incidents over a 13-year timeframe from November 2005 to October 2018“
- step break-out
the term step is not commonly used in rock climbing discriptions. The correct term should be foot hold.
Although spell checked and corrected by an English native speaking official court translator, we must completely agree on your request!
We changed step to foothold within the complete manuscript.
- undesirable outcome
what is an undesirable outcome, a fatality? A permanent disability?
We must agree that this phrase can be misleading. Our definition of undesirable outcome (as stated later on) was a severe or fatal injury. For better understanding, particularly within the abstract we decided to change the sentence as follows:
Line 27: “The risk for severe or fatal injuries increased with age and fall height, when ascending or bouldering, during the morning hours and when climbing without a helmet or rope“
In results no injury grading is presented. Please present grading according to a common used classification, best the UIAA medcom classification.
Dear Reviewer, we completely understand your objection!
Injury severity was documented by alpine police officers by utilizing the definition given by Austrian law. Unfortunately, clinical in-hospital data were missing. We decided to translate a severe injury (defined as: any fracture, except a fracture of the nose, or the patient is unable to work for at least 24 days after the accident) into a UIAA MedCom category 3-5. A mild injury was therefore categorised as UIAA MedCom 1-2 and uninjured as UIAA MedCom 0.
The following line in the abstract was changed:
Line 23: “Acute traumatic injuries (n=1,217) were often classified as severe (UIAA≥3; n=709) and commonly involved fractures (n=566).“
Conclusion:
- Acute traumatic injuries when
- 30 rock climbing are often accompanied by poor outcome in terms of severe or fatal injuries.
No, most acute injuries in all other studies are not severe, while severe injuries are still happening. Nevertheless most injuries are ankle fractures etc and not severe.
Dear Reviewer,
We do understand your concerns with our data. Importantly one must differentiate between acute traumatic, atraumatic and chronic overuse injuries. To further emphasize this point we decided to add the following to the introduction part:
Line 122: “When analyzing rock climbing related injury patterns, differentiation between acute traumatic (e.g. fall from height, rockfall), atraumatic (supraphysiological loading) and chronic overuse injuries (e.g. pulley strain) seems wise.“
Of course, atraumatic and chronic overuse injuries mostly reported on cannot be seen as severe! Whether acute traumatic injuries are classified as severe or not, mainly depends on the definition of a severe injury and the population analyzed. In our case the sustained injuries were mainly classified as severe. But in our case our study population was also defined by being involved in an emergency and requiring professional rescue!
To better clarify this point we rephrased the sentence as follows:
Line 30: “Acute traumatic injuries requiring professional rescue when rock climbing are often severe or fatal.“
Also, in the Discussion (4.2. Injury severity and localization) we added the following:
Line 501: ” With regards to the rescue data presented in this study, we must assume that victims with milder and/or upper limb injuries were rather capable of self-rescue than those with more severe injuries or lower limb affection. This is supported by the high proportion of fractures, lower limb involvement and severely injured reported by us and is in line with other comparable studies.[21]”
As mentioned in this sentence. Our data are in line with other comparable studies (Lack et al from Boulder County).
Introduction
- The existing literature mainly focuses on injuries reported
- 46 in questionnaires or on climbers treated in an emergency department.[2–14] In both cases overuse
- 47 injuries caused by an excessive climbing load are predominantly addressed.
This statement is false for the studies involving Emergerncy departements only, eg. Buzzacot and Nelson. These NEISS data exclude overuse injuries.
Dear Reviewer,
Thank you once more for your very attentive reading!
We must agree to your objection and changed the sentence as follows:
Line128: “A bulk of recent literature mainly focuses on injuries reported in questionnaires or on climbers treated in an emergency department.[3,7,9–19] As acute traumatic injuries infrequently occur, literature is rather scarce and the more common overuse injuries are predominantly addressed.“
- While overuse injuries
- 48 are often related to forced and prolonged training intensity,
should be training and climbing intensity
Yes! Fully agree. Sentence is changed to the following:
Line 131 :” While overuse injuries are often related to forced and prolonged training and climbing intensity, accidents are mainly caused by errors, inattentiveness or environmental influences.[3]”
- A study from Grand Teton National Park, WY, USA, reported 144 rock climbing accidents in a 10-
- 53 year period from 1970 to 1980.[15]
This study is good, but very old. The climbing techniques in the 80ies vary from today, please address this.
We added the following sentence to address this issue:
Line 138: “When referencing to this study one must keep in mind that climbing techniques differed substantially compared to now“
Methods
- Severity of injury was defined according to Austrian law: an injury was considered mild if the
- 86 patient was able to resume work within 24 days and severe if any fracture, except a fracture of the
- 87 nose, occurred or the patient was unable to work for at least 24 days after the accident.[19
I would strongly suggest to use a standard classification of trauma severity. I recommend the UIAA medical commission score as the int. climbing medicine bodies classification.
Dear Reviewer,
As mentioned above we truly do understand your objection and added the following to the methods section:
Line 179: “Due to regulations provided by Austrian law, documentation of injury severity initially took place according to the following definition: an injury is considered mild if the patient is able to resume work within 24 days and severe if any fracture, except a fracture of the nose, occurrs or the patient is unable to work for at least 24 days after the accident.[23] Aiming for comparability we translated this definition into the following UIAA MedCom Scores: uninjured – 0; mild injury – 1-2; severe injury – 3-5; death on-site – 6.[24] As in-hospital data was missing to verify or further classify injury severities a more precise grading was not possible.“
In Table 2 the following was added:
Injury severity |
Uninjured |
UIAA MedCom 0 |
1469 (49.1) |
- step break-out
please check with a native speaker if this term is correct. I am not a native English speaker but it seems unusual to me and I know the term foothold is mostly used.
As mentioned above we decided to change this phrasing within the complete manuscript!
The phrase is now: “foothold break-out”.
Results
Table 2 : exhaustion is not an injury but a medical condition, same for hypothermia. You may need to rephrase injury into injuries and medical conditions, same for stroke etc.
We fully agree and changed the subheading to “Emergency classification” in Table 2.
Injury localization (1503 entries in n= 1315 patients)
Also here we would recoomend to use a classification, e.g. OSICS for localization.
Dear Reviewer,
Once more we truly understand your objection. Regarding injury localization it was only possible for us to work with the data we had, hence the data documented. We do feel though, that documentation quality was high and localizations represented well!
Regarding numbers (if that was a question): When more than one localization was affected, more than one localization was documented per patient. E.g. ankle fracture and laceration to the arm…
Regarding classification: nearly all OSICS localizations are represented. Merely spine injuries were not further classified into cervical, thoracal or lumbar spine!
Table 3
18 – 29 yrs.# < 18 yrs.
Why don’t you start chronologically, with under 18 then 18-256 etc..?
Very good and attentive question!
The main reason we decided to set this age group as reference is that the youngest age group (<18 yrs) consisted only of 93 persons and therefore was a very small group!
We feared the model could become incorrect and therefore referenced our odds ratios toward the second youngest and also second largest age group (18 – 29 yrs.)!
Climbing discipline:
How do you differ from alpine to sport? Please define
Dear Reviewer,
this issue was brought up by another Reviewer as well! We decided on adding a complete paragraph to the introduction:
Line 39fff: “The popularity of rock climbing has grown steadily since it grew out of mountaineering during the last century. Since then, rock climbing has split into several categories, namely alpine (traditional) climbing, ice climbing, sport climbing and bouldering. Briefly describing these categories, alpine climbing is performed by placing and removing all gear necessary (e.g. nuts, cams, slings) for ascent and also fall protection into a rock wall yourself. Ice climbing can be seen as a special subcategory of alpine climbing on inclined ice formations and extreme variants, such as free solo climbing where climbers climb alone without belaying and aiding equipment also. Sport climbing and bouldering however can be performed in- and outdoors. In outdoor sport climbing permanent anchors and bolts are prefixed in close distance to the wall suggesting a higher protection when used correctly. Additional protection can be provided by the use of crash pads for falls from short distance. These pads are particularly useful when bouldering, which is performed without any equipment on small rock formations outdoors or on artificial walls in low height indoors.“
- Helmets were usually worn and ropes usually used when involved in a rock climbing
- 154 emergency (Table 3).
also in sport climbing??
Dear Reviewer,
Again, we would like to thank you for a very valuable hint!
First, we noticed that there might be potential for misunderstanding and found it important to discriminate that we are solely talking about outdoor sport climbing in this study and added this prefix were appropriate.
Second, we analysed helmet use in special regard to sport climbing. Within our data the sentence is true. Particularly with regards to sport climbing it actually isn’t (same for bouldering of course).
We changed the following sentences as follows:
Within the results we added “in general”
Line 399: “In general, helmets were usually worn and ropes usually used when involved in a rock climbing emergency (Table 3).“
And within the Discussion:
Line 545ff: „. Particularly regarding outdoor sport climbing 58% were not wearing a helmet prior to the emergency. Of them, 43% were injured severely and 3% fatally. The present data remind us that consistent use of a helmet also when performing outdoor sport climbing and acute vigilance when belaying can help prevent an unfortunate outcome.“
Discussion
- Consistent use of a helmet, also when sport climbing,
- 173 consistent use of a rope, particularly on an ascent, high vigilance while belaying, including proper
- 174 training in belaying, thorough tour preparation and situational awareness, especially with regard to
- 175 personal fitness and environmental conditions, can help prevent poor outcome.
even if logical it can be stated for sure as not proofed scientifically. I would suggest change to are likely or may protect
We must completely agree!
Also due to additional comments by other reviewers we rephrased this conclusion to the following:
Line 423: “Consistent use of a helmet, also when sport climbing, consistent use of a rope, particularly on a traditional ascent, proper training in and high vigilance while belaying, proper spotting when bouldering, situational awareness, especially with regard to personal fitness and environmental conditions are likely to help prevent poor outcome“
Also in Line 620 (Conclusions):
„Consequent use of a helmet also when sport climbing, consistent use of a rope, particularly when ascending, proper spotting when bouldering and proper training as well as high vigilance when belaying may help prevent such injuries.“
Also in the Abstract Line 30: „Consequent use of a helmet also when sport climbing, consistent use of a rope, particularly when ascending, proper spotting when bouldering and proper training as well as high vigilance when belaying are likely to help prevent such injuries.“
- The use of a helmet is known to prevent undesirable outcomes, not only in rock climbing.
- 216 Appalling enough, that in certain areas up to 80% of climbers practice rock climbing without a
- 217 [20,23] Only 18% of all cases in this study involved persons not wearing a helmet, but the
- 218 percentage increased to 28% in the group with undesirable outcome. The present data remind us that
- 219 consistent use of a helmet also when performing sport climbing and acute vigilance when belaying
- 220 can help prevent an unfortunate outcome.
well I certainly disagree for sport climbing and other studies on sport climbing injuries showed very little head injuries… this needs to be discussed.
As mentioned above, we must emphasize that our data represent an analysis of rock climbing emergencies requiring professional rescue involving alpine police officers. Hence, our data consists of outdoor mountainous rock climbing emergencies. With regard to sport climbing, only outdoor sport climbing was considered! Of course, also while performing sport climbing outdoors, athletes can be affected by rockfall or falls with rock impact. Therefore, we truly believe that, in the described setting head injuries can be likely.
Our data does show that helmet use was more seldom in sport climbing:
Discussion Line 545ff: „. Particularly regarding outdoor sport climbing 58% were not wearing a helmet prior to the emergency. Of them, 43% were injured severely and 3% fatally. The present data remind us that consistent use of a helmet also when performing outdoor sport climbing and acute vigilance when belaying can help prevent an unfortunate outcome.“
But our data also shows that within the group not wearing a helmet but involved in emergencies requiring professional rescue and injured severely or fatally ascending while sport climbing was the most frequently performed discipline and activity (seen in Appendix Table 1).
To clarify this issue, we did add the following to the first paragraph of the discussion:
Line 418ff: “With regard to rock climbing related incidents requiring professional rescue, odds ratios for severe or fatal injuries increased with age, height of fall, while bouldering or ascending, during the morning hours and when climbing without a helmet or rope. Particularly with regard to the latter two, severe or fatal injuries while climbing without a helmet or rope mainly occurred while ascending in sport climbing concerning helmet and while ascending in alpine climbing concerning rope usage. Consistent use of a helmet, also when sport climbing, consistent use of a rope, particularly on a traditional ascent, proper training in and high vigilance while belaying, proper spotting when bouldering, situational awareness, especially with regard to personal fitness and environmental conditions are likely to help prevent poor outcome.“
- Spring was the season with the highest rate of severe or fatal injuries. The most plausible
- 234 explanation for this phenomenon is that overall fitness was probably somewhat poorer in spring than
- 235 in the summer or fall.
also: in winter rock freezes, then I taws, and rock fall is always higher in spring!
Dear Reviewer,
Thank you for your valuable input. Due to recommendations by another reviewer we changed the confidence interval of odds ratios in our logistic regression analysis. Due to this, seasonal differences turned out non-significant. We decided to remove this part from the results and discussion.
Tables and Figures
Conclusion:
- Conclusions
- 276 In rock climbing accidents requiring professional rescue, the main causes of death and severe
- 277 injuries were falls after a stumble, grip or step break-out or a belaying error. Consequences were
- 278 multiple trauma or head injuries in the fatal cases and lower extremity injuries in the severely injured.
- 279 The risk for such undesirable outcomes increased with age and height of fall, when ascending or
- 280 bouldering, in spring, during the morning hours and when climbing without a helmet or rope. The
- 281 case fatality rate was 4.7% and overall mortality was estimated at 0.003–0.007 per 1000 h of rock
- 282 Preventive measures can be consistent use of a helmet also when sport climbing, consistent
- 283 rope use particularly when ascending, acute vigilance when belaying and proper belay training,
- 284 thorough tour preparation and situational awareness, especially concerning personal fitness and
- 285 environmental conditions. While bouldering, spotting the climber and correct positioning of high-
- 286 quality crash pads may help to reduce injuries.
this is the same as the abstract! Should actually draw a conclusion, be more precise and shorter
Once more we must agree to your objection! We changed the conclusion as follows:
Line 618: “Although general mortality is low and the risk being in need for rescue from a mountainous environment uninjured high, acute traumatic injuries requiring professional rescue when rock climbing are often severe or fatal. Consequent use of a helmet also when sport climbing, consistent use of a rope, particularly when ascending, proper spotting when bouldering and proper training as well as high vigilance when belaying may help prevent such injuries.”
Round 2
Reviewer 1 Report
Authors addressed all my comments. I recommend accept this manuscript for publication.
Reviewer 3 Report
all points are addressed appropriate